# A deep learning reconstruction of mass balance series for all glaciers in the French Alps: 1967-2015

Jordi Bolibar[1,2], Antoine Rabatel[1], Isabelle Gouttevin[3], and Clovis Galiez[4]

[1]Univ. Grenoble Alpes, CNRS, IRD, G-INP, Institut des Géosciences de l'Environnement (IGE, UMR 5001), Grenoble, France
[2]INRAE, UR RiverLy, Lyon-Villeurbanne, France
[3]Univ. Grenoble Alpes, Université de Toulouse, Météo-France, CNRS, CNRM, Centre d'Études de la Neige, Grenoble, France
[4]Univ. Grenoble Alpes, CNRS, G-INP, LJK, Grenoble, France

**Correspondence:** Jordi Bolibar (jordi.bolibar@univ-grenoble-alpes.fr)

**Abstract.** Glacier mass balance (MB) data are crucial to understand and quantify the regional effects of climate on glaciers and the high-mountain water cycle, yet observations cover only a small fraction of glaciers in the world. We present a dataset of annual glacier-wide mass balance of all the glaciers in the French Alps for the 1967-2015 period. This dataset has been reconstructed using deep learning (i.e. a deep artificial neural network), based on direct MB observations and remote sensing annual estimates, meteorological reanalyses and topographical data from glacier inventories. The method's validity was assessed previously through an extensive cross-validation against a dataset of 32 glaciers , with an estimated average error (RMSE) of 0.55 m.w.e. $a^{-1}$, an explained variance ($r^2$) of 75% and an average bias of -0.021 m.w.e. $a^{-1}$. We estimate an average regional area-weighted glacier-wide MB of -0.69±0.21 ($1\sigma$) m.w.e. $a^{-1}$ for the 1967-2015 period, with negative mass balances in the 1970s (-0.44 m.w.e. $a^{-1}$), moderately negative in the 1980s (-0.16 m.w.e. $a^{-1}$), and an increasing negative trend from the 1990s onwards, up to -1.26 m.w.e. $a^{-1}$ in the 2010s. Following a topographical and regional analysis, we estimate that the massifs with the highest mass losses for the 1967-2015 period are the Chablais (-0.93 m.w.e. $a^{-1}$), Champsaur (-0.86 m.w.e. $a^{-1}$) and Haute-Maurienne and Ubaye ranges (-0.84 m.w.e. $a^{-1}$ both), and the ones presenting the lowest mass losses are the Mont-Blanc (-0.68 m.w.e. $a^{-1}$), Oisans and Haute-Tarentaise ranges (-0.75 m.w.e. $a^{-1}$ both). This dataset - available at: https://doi.org/10.5281/zenodo.3925378 (Bolibar et al., 2020a) - provides relevant and timely data for studies in the fields of glaciology, hydrology and ecology in the French Alps, in need of regional or glacier-specific annual net glacier mass changes in glacierized catchments.

## 1 Introduction

Among all the components of the Earth system, glaciers are some of the most visibly affected by climate change, with an overall worldwide shrinkage despite important differences between regions (Zemp et al., 2019). The European Alps are among the regions with the strongest glacier mass loss over recent decades, with expected mass losses between 60% and 95% by the end of the 21st century (Zekollari et al., 2019). These major glacier mass changes are likely to have an impact on water

resources, society and alpine ecosystems (e.g. Huss and Hock, 2018; Immerzeel et al., 2020; Cauvy-Fraunié and Dangles, 2019). In order to study and quantify all these potential consequences, the availability of glacier mass balance data is of high relevance. Therefore, open historical datasets are crucial for the understanding of the driving processes and the calibration of models used for projections. Unlike glacier length, glacier mass balance (MB) provides a more direct indicator of the climate-glacier interactions (Marzeion et al., 2012). Glacier surface mass balance (SMB) is classically measured using the direct or glaciological method, by separately determining the ablation and accumulation totals. Direct measurements quantify the surface mass balance at different points of the glacier, and these values must be integrated at the glacier scale in order to assess the glacier-wide SMB (Benn and Evans, 2014). These different point SMB measurements can show a high nonlinear variability, which can complicate this integration process towards glacier-wide estimates (Vincent et al., 2018). Moreover, field measurements require a lot of manpower, time and economic resources in order to be sustained for a meaningful period of time. On the other hand, recent advances in remote sensing allow estimating glacier MB changes at a regional level with unprecedented efficiency using geodetic and gravimetric methods (Kääb et al., 2012; Fischer et al., 2015; Berthier et al., 2016; Brun et al., 2017; Dussaillant et al., 2019). Due to constraints related to the availability of digital elevation models (DEMs) or airborne data, these mass balance estimates normally encompass several years or decades. Some studies are bridging the gap towards an annual temporal resolution (Rabatel et al., 2005, 2016; Rastner et al., 2019), but the coverage is still limited to glaciers without cloud cover or acquisition-related artefacts. This means that these mass balance datasets are often restricted to certain glaciers and years within a region. All these new datasets are extremely beneficial for data-driven approaches, fostering the training of machine learning models capable of capturing the regional characteristics and relationships (Bolibar et al., 2020b). This type of approach allows to fill the spatiotemporal gaps in the MB datasets, therefore, it can be seen as a complement to remote sensing and direct observations.

On the other hand, MB reconstructions have already been carried out in the European Alps, providing a basis for comparison between different approaches (see Hock et al. (2019) for a compilation). Two studies include reconstructions in the European Alps, including the French Alps, over a substantial period of the recent past: Marzeion et al. (2012, 2015) reconstructed annual MB series of all glaciers in the Randolph Glacier Inventory for the last century. They used a minimal model relying only on temperature and precipitation data, based on a temperature-index method, with two parameters to calibrate the temperature sensitivity and the precipitation lapse rate. Huss (2012) presented an approach to extrapolate SMB series of a limited number of glaciers to the mountain-range scale. By comparing multiple methods, he found the best results with a multiple linear regression based on 6 topographical parameters. From this relationship he reconstructed area-averaged SMB series of all the glaciers of the European Alps between 1900-2100 and analysed the trends for the different alpine nations and different glacier sizes.

Here, we introduce a dataset of annual glacier-wide MB of all the glaciers in the French Alps (Bolibar et al., 2020a), located in the westernmost part of the European Alps, between 5.08° and 7.67°E, and 44° and 46°13'N. Glacier-wide MBs have been reconstructed for the 1967-2015 period, using deep learning (i.e. a deep artificial neural network) (Fig. 1). This approach was introduced in Bolibar et al. (2020b), for which a deep artificial neural network (ANN) was trained with data from 32 French alpine glaciers, as part of the ALpine Parametrized Glacier Model (ALPGM) (Bolibar, 2020). Annual glacier-wide MB values

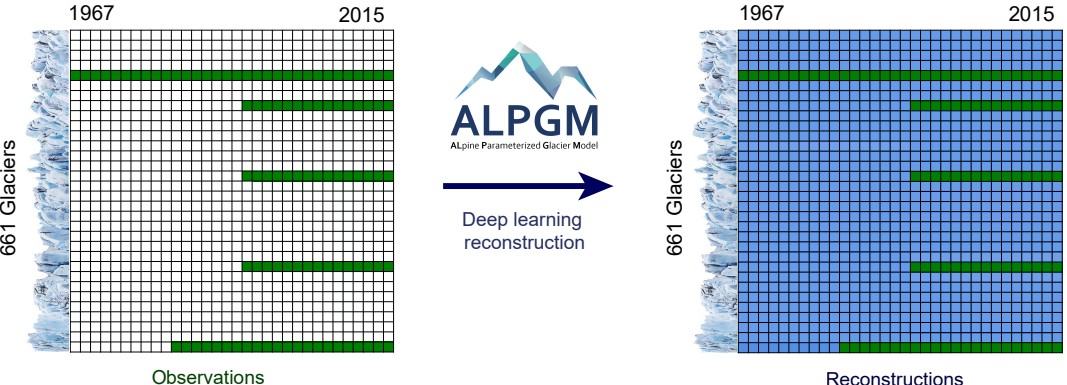

**Figure 1.** Summary of the deep learning regional MB reconstruction approach. From the available annual glacier-wide MB data, a deep learning model is used to reconstruct the full dataset, thus filling the spatiotemporal gaps in the observational dataset. Green indicates glaciers and years with MB observations and remote sensing estimates, and blue indicates reconstructed MB values. Glacier ice cliffs in the vertical axis indicate rows representing individual glaciers. The grid size with glaciers and years is schematic and only serves to illustrate the concept.

are reported for each glacier in the French Alps found in the 2003 glacier inventory (Gardent et al., 2014). An overview of the methodology used to produce the dataset and a review of the associated uncertainties is presented in Sect. 2, followed by a dataset overview in Sect. 3, where the data structure and regional trends are described and where the dataset is compared to a previous study and observations.

## 2  Data and methods

### 2.1  Training data

For the reconstruction presented here, a dataset of 32 French alpine glaciers has been used for training, covering most of the massifs within the French Alps, which exhibit a great variability of topographical characteristics (Fig. S10). The French Alps are located in the westernmost part of the European Alps, rising from the Mediterranean sea northwards between 44 and 46°13' N, 5.08 and 7.67° E. Due to its particular geographical setup, glacierized mountain ranges in the French Alps have distinct climatic signatures. Southern glaciers exhibit a Mediterranean influence, whereas northern glaciers are mostly affected by western fluxes from the Atlantic, except for eastern glaciers close to the Italian border, which are more influenced by east returns.

Out of the 32 glaciers from this dataset, four glaciers include direct MB measurements from the GLACIOCLIM observatory, some of which since 1949. These direct observations have been calibrated using photogrammetric geodetic MB (Vincent et al., 2017). On the other hand, 28 glaciers include estimates of annual glacier-wide MB from remote sensing between 1984 and

2014 (Rabatel et al., 2016). These remote sensing estimates were computed using (1) the end-of-summer snowline for every year, which in the European Alps is a proxy of the equilibrium-line altitude (ELA); and (2) geodetic MB for the 1984-2014 period quantified from two high-resolution DEMs. Both data sources are used to reconstruct the annual glacier-wide MB of each individual glacier for the same period of the geodetic MB.

This dataset of 32 glaciers, with a total of 1048 annual glacier-wide MB values, is used as a reference. Unlike point MB, glacier-wide MB is influenced by both climate and glacier geometry, producing complex interactions between climate and glacier morphology that need to be taken into account in the model. For each annual glacier-wide MB value available, the following data are compiled to train the ANN with an annual time step: (1) climate data from the SAFRAN meteorological re-analyses (Durand et al., 2009), with: cumulative positive degree days (CPDD), cumulative winter snowfall, cumulative summer

snowfall, mean monthly temperature and mean monthly snowfall, all variables being quantified at the altitude of the glacier's centroid. In order to capture the climate signal at each glacier's centroid, temperatures are taken from the nearest SAFRAN 300 m altitudinal band and adjusted with a 6 ºC/km lapse rate. The updated temperature is then used to update the rain-snow parts from the same 300 m altitudinal band. Snowfall is considered as all precipitation fallen at temperatures equal or lower than 0º C. (2) annually interpolated topographical data between the 1967, 1985, 2003 and 2015 glacier inventories in the French

Alps (update of Gardent et al., 2014), with: mean and maximum glacier altitude, slope of the lowermost 20% altitudinal range of the glacier, surface area, latitude, longitude and aspect. Therefore, the topographical feedback of the shrinking glaciers is captured from these annually interpolated topographical predictors. These topoclimatic parameters were identified as relevant for glacier-wide MB modelling in the French Alps (Bolibar et al., 2020b), and the dates of the glacier inventories determined the time interval for the reconstructions presented here.

For more details on the choice of predictors, the reader can find a more detailed analysis in Bolibar et al. (2020b).

## 2.2  Methods

The annual glacier-wide MB dataset for the 661 French alpine glaciers has been reconstructed using a deep artificial neural network (ANN), also known as deep learning. ANNs are nonlinear statistical models inspired by biological neural networks (Fausett, 1994; Hastie et al., 2009). Recent developments in the field of machine learning and optimization enabled the use of

deeper ANN architectures, which allows capturing more nonlinear and complex patterns in data even for small datasets (Ingrassia and Morlini, 2005). This modelling approach is part of the MB component of ALPGM (Bolibar, 2020), an open-source data-driven parameterized glacier evolution model. For a detailed explanation of the methodology, please refer to Bolibar et al. (2020b). For the final reconstructions presented here, a cross-validation ensemble approach was used based on 60 Leave-Some-Years-and-Glaciers-Out (LSYGO) cross-validation models. Individual predictions of each of the members were averaged to

produce a single output. An ensemble approach has the advantage of further improving generalization, and reducing overfitting as well as the inter-model high variance typical from neural networks (Krogh and Vedelsby, 1995). A weighted bagging approach (Hastie et al., 2009) was used in order to balance the dataset, giving more weight to under-represented data samples from the years 1967-1983. On the other hand, for the 32 glaciers with glacier-wide MB observations and remote sensing estimates used for training, an ensemble of 50 models trained with the full dataset was used, in order to achieve the best possible

performance for this subset of glaciers, which represents a substantial fraction (45% in 2003) of the total glacierized surface area in the French Alps.

## 2.3 Uncertainty assessment

The uncertainties linked to the deep learning approach used in this study have been assessed through cross-validation, for which deep learning predictions were compared with observations and remote sensing estimates. A detailed presentation of the method's uncertainties and performance from the cross-validation study can be found in Bolibar et al. (2020b). Block cross-validation ensured that all the 32 glaciers in the dataset were evaluated, with spatiotemporal structures formed by glaciers and years being considered in order to prevent the violation of the assumption of independence (Roberts et al., 2017). This means that three different deep ANNs were produced: one for reconstructing glacier-wide MB in space, one for the reconstruction in time (future and past), and another one for both dimensions at the same time; each of these with a different calibration and performance. It was shown that the deep ANN performs better in the spatial dimension, in which the MB signal relationships with the predictors are the simplest. MB annual variability is mostly driven by climate, whereas geography and local topography (i.e. differences between glaciers) modulate the signal in space in a simpler way (Vincent et al., 2017; Bolibar et al., 2020b). Therefore, deep learning is capable of finding more structures in the spatial dimension, accounting for a better accuracy and explained variance compared to the temporal dimension. The deep ANN used in this study presents an RMSE of 0.55 m.w.e a$^{-1}$ with an $r^2$ of 0.75 in LSYGO cross validation. The ANN MB reconstructions accurately reproduce the annual variability of glaciological observations from the GLACIOCLIM observatory (Figure S1). This reinforces the trust in the produced model ensemble, indicating that models trained with heterogeneous data comprised by glaciological and remote sensing estimates can correctly reproduce direct annual observations.

Nonetheless, only one glacier in the training dataset is smaller than 0.5 km$^2$ (Glacier de Sarennes, 0.3 km$^2$ in 2003), implying that uncertainties for very small glaciers ($< 0.5$ km$^2$) might differ from those estimated using cross-validation. In 2015, very small glaciers in the French Alps represented about 80% of the total glacier number, but they accounted for only 20% of the total glacierized area. This means that their importance is relative, for example in terms of water resources, but a user of this dataset should bear in mind that MB from these very small glaciers might carry greater uncertainties than the ones assessed during cross-validation. This might be especially true for extremely small glaciers ($< 0.05$ km$^2$) which can be considered as spatial outliers for the deep ANN. Since there is only one glacier with MB observations for very small glaciers and none for extremely small glaciers, there is no precise way to quantify these uncertainties. On the other hand, the ANN is mostly trained with glacier-wide MB data between 1984 and 2014, with a reduced amount of values between 1967 and 1984 (986 and 62 values, respectively). Since this early period contains on average more positive and neutral glacier-wide MB values than the 1984-2014 period, the performance of the ANN was specifically assessed for this period. An additional cross-validation was performed with four folds, each with a glacier including glacier-wide MB data before 1984. For each fold, all MB data of that glacier and time period were hidden from the ANN, and the simulated glacier-wide MBs between 1967 and 1983 were tested in order to assess the model's performance. The results showed that the ANN is capable of correctly reconstructing glacier-wide MB for glaciers and years before 1984 (Fig. S5), with an estimated accuracy (RMSE) of 0.47 m.w.e. a$^{-1}$ and an estimated

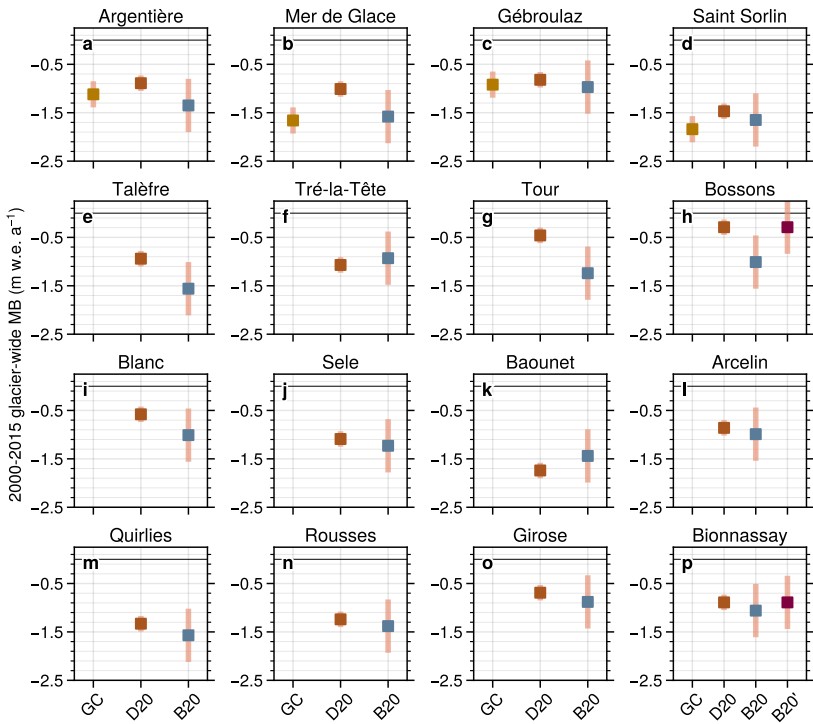

**Figure 2.** Comparison of average annual glacier-wide MB for the 2000-2015 period between the glaciological MB from the GLACIOCLIM observatory (GC), the ASTER-derived geodetic MB from Davaze et al., 2020 (D20), the MB reconstructions from this study (B20) and the reconstructions from this study recalibrated using the ASTER-derived geodetic MB (B20').

explained variance ($r^2$) of 0.65. This uncertainty assessment is based on roughly 10% of the full dataset, meaning that these estimates lack the robustness of the full cross-validation from Bolibar et al. (2020b), but they serve to show that the model can accurately reconstruct glacier-wide MB data outside the main cluster of years used during training.

In order to further validate the reconstructions presented here, a comparison against independent ASTER (Davaze et al., 2020) and Pléiades (Berthier et al., 2014) geodetic MB data was performed, that helps to assess the bias of the MB reconstructions for the 2000-2015 (Fig. 2) and 2003-2012 (Fig. S2) sub-periods. The photogrammetric geodetic MB used to calibrate the MB datasets from Rabatel et al. (2016) and the glaciological observations from GLACIOCLIM have a much higher resolution than ASTER-derived geodetic MB, but the comparison can bring interesting information for glaciers outside the training dataset. Our reconstructions show a good agreement with the geodetic MB for certain regions (e.g. Grandes Rousses), except for some particular steep large high-altitude glaciers (e.g. Bossons and Taconnaz in the Mont-Blanc massif) that substantially differ from most glaciers in the French Alps. A more detailed analysis and additional figures comparing the MB datasets can be found in Sect. 1 of the Supplementary. In order to exploit this additional geodetic MB dataset, we have recalibrated our MB reconstructions for the 2000-2015 period using the ASTER-derived geodetic MB from Davaze et al. (2020) for some glaciers

outside our training dataset (i.e. B20' in Fig. 2). Since ASTER-derived geodetic MB present important uncertainties for small glaciers (i.e. $< 1$ km$^2$), we have only recalibrated MB series for 16 large glaciers outside the training dataset with uncertainties lower than 0.15 m.w.e. a$^{-1}$. The calibration has been performed by adding the average annual bias between Davaze et al. (2020) and this study for the 2000-2015 sub-period.

## 3   Dataset overview

### 3.1   Dataset format and content

The MB dataset is presented in two different formats: (a) A single netCDF file containing the MB reconstructions, the glacier RGI and GLIMS IDs and the glacier names. This file contains all the necessary information to correctly interact with the data, including some metadata with the authorship and data units. (b) A dataset comprised of multiple CSV files, one for each of the 661 glaciers from the 2003 glacier inventory (Gardent et al., 2014), named with its GLIMS ID and RGI ID with the following format: *GLIMS-ID_RGI-ID_SMB.csv*. Both indexes are used since some glaciers that split into multiple sub-glaciers do not have an RGI ID. Split glaciers have the GLIMS ID of their "parent" glacier and an RGI ID equal to 0. Every file contains one column for the year number between 1967 and 2015 and another column for the annual glacier-wide MB time series. Glaciers with remote sensing-derived estimates (Rabatel et al., 2016) include this information as an additional column. This allows the user to choose the source of data, with remote sensing data having lower uncertainties (0.35±0.06 ($\sigma$) m.w.e. a$^{-1}$ as estimated in Rabatel et al. (2016)). Columns are separated by semicolon (;). All topographical data for the 661 glaciers can be found in the updated version of the 2003 glacier inventory included in the Supplementary material and in the dataset repository.

### 3.2   Overall trends

We estimate an average area-weighted regional glacier-wide MB of -0.69±0.21 ($\sigma$) m.w.e. a$^{-1}$ between 1967 and 2015 (Fig. 3 and 4). As reported in previous studies (Huss, 2012; Rabatel et al., 2016; Vincent et al., 2017), our reconstructed MB data show a slightly negative average value during the 1970s, even less negative in the 1980s, and then increasingly negative values in recent decades with an abrupt change in 2003 (Fig. 2). For this period (1967-2015), the year 2003 with its remarkable heatwave remains the most negative glacier-wide MB year (-2.26 m.w.e. a$^{-1}$ on average), with 1984 being the most positive year of the study period (+0.85 m.w.e. a$^{-1}$ on average). The area-weighted average MB is slightly less negative than the mean annual glacier-wide MB, showing a light asymmetry in the probability distribution function (PDF) (Fig. 3c).

### 3.3   Regional and topographical trends

Here we analyse the main trends for the glacierized massifs and for some relevant topographical parameters. The reported glacier-wide MBs are only area-weighted if specifically mentioned. Interesting differences appear once the dataset is divided into mountain ranges (Fig. 5). The Mont-Blanc massif presents the lowest mass loss over the entire study period, with an average cumulative loss over the 1967-2015 period of 33.5 m.w.e. This is probably due to its northern location within the

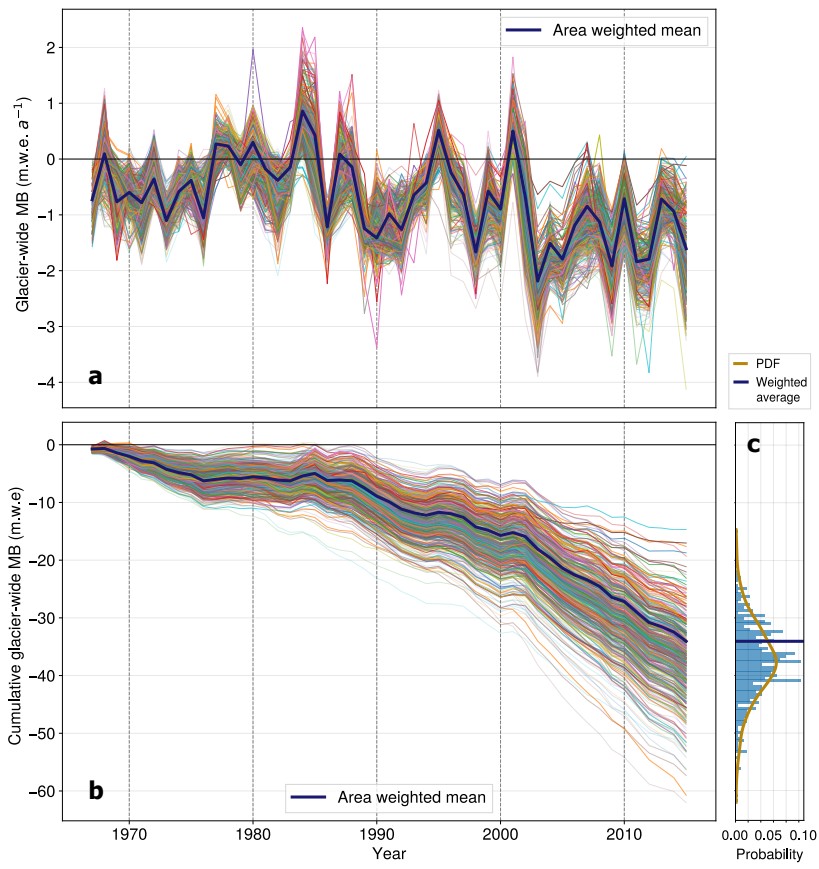

**Figure 3.** (a) Annual glacier-wide MB and (b) cumulative glacier-wide MB reconstructions of all the glaciers in the French Alps ($N = 661$) between 1967 and 2015. For each individual glacier, line thickness depends on glacier area, with smaller glaciers having thinner lines. The histogram (c) indicates the distribution and probability density function (PDF) of the 1967-2015 cumulative MB (m w.e.) of the dataset.

French Alps and its large high altitude accumulation areas, which resulted in more positive or less negative MBs, especially during the 1980-2000s. Oisans is the massif with the second lowest average cumulative mass loss (37.20 m.w.e.). Its glaciers have average altitudes ranging from 2290 to 3470 m.a.s.l., with around 50% of them having mean altitudes over 3000 m.a.s.l. and with about 40% of glaciers (including most of the large ones) having a northern aspect. Glaciers in Haute-Tarentaise

5  present similar characteristics to those from Oisans, with mean altitudes ranging between 2300 and 3600 m.a.s.l., with about 60% of the glaciers above 3000 m.a.s.l. This less negative trend was especially important during the recent years with high mass losses from 2003 onwards. On the other hand, the Ubaye, Champsaur, Chablais and Haute-Maurienne massifs appear as the most affected mountain ranges with cumulative mass losses reaching between 41 and 46 m.w.e. for the four massifs over the 1967-2015 period. The Chablais range has a very small number of glaciers remaining, all of them at rather low altitudes (2200-

10  2900 m.a.s.l.), relatively small (0.01 - 1.1 km²), and with a northwestern aspect. Despite being the northernmost mountain

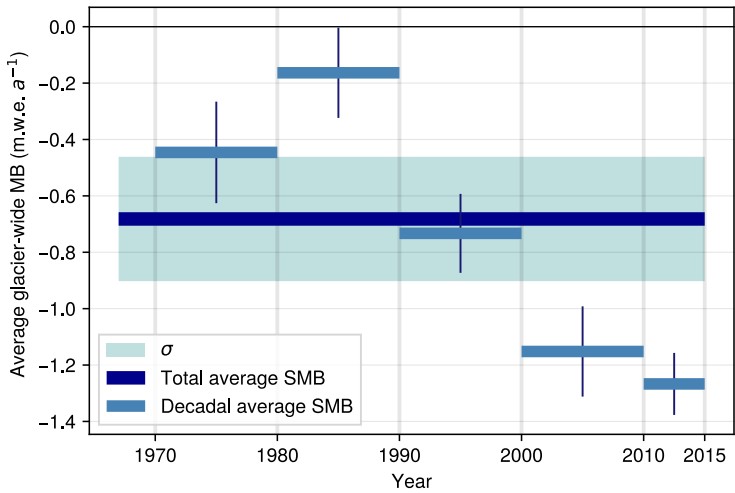

**Figure 4.** Averaged area-weighted decadal glacier-wide MB for the French Alps with decadal uncertainties. The total area-weighted glacier-wide MB is estimated for the 1967-2015 period.

range in the French Alps, its low altitude is most likely the main reason for the very negative MBs, which were under the regional average even during the positive years in the 1980s. The Champsaur range shows a similar situation, with very small glaciers (0.03 - 0.89 km$^2$) lying at relatively low altitudes (2300-3100 m.a.s.l.) in the southernmost latitudes of the Alps (44°7'). Finally, the situation of the Ubaye massif is quite similar to the one of Champsaur, being the southernmost glacierized massif

in the French Alps, with a strong mediterranean influence. Such glaciers are remnants of the Little Ice Age, far from being in equilibrium with the warming climate, and can quickly lose a lot of mass through non-dynamic downwasting (Paul et al., 2004).

When classifying the MB time series by glacier surface area, we encounter the following patterns, with $n$ being the number of glaciers in the subset and $s$ its standard deviation: (1) Very small glaciers ($< 0.5$ km$^2$; $n = 534$; $\overline{MB}_{1967-2015}$ = -0.79 m.w.e.

a$^{-1}$; $s = 0.23$ m.w.e. a$^{-1}$) present more negative glacier-wide MBs than (2) small/medium glaciers (ranging from 0.5 to 2 km$^2$; $n = 93$; $\overline{MB}_{1967-2015}$ = -0.74 m.w.e. a$^{-1}$; $s = 0.18$ m.w.e. a$^{-1}$) and (3) large glaciers ($> 2$ km$^2$; $n = 34$; $\overline{MB}_{1967-2015}$ = -0.68 m.w.e. a$^{-1}$; $s = 0.14$ m.w.e. a$^{-1}$) (Fig. S8). Very small glaciers present a larger spread of values than small/medium and large glaciers ($s = 0.23$ m.w.e. a$^{-1}$ versus 0.18 and 0.14 m.w.e. a$^{-1}$, respectively). As explained in Sect. 2, the uncertainties for very small glaciers are greater due to their under-representation in the training dataset, meaning that analyses based on

small glaciers have to be taken with greater care. The effects of these trends can be seen in the PDF of the cumulative MB reconstructions (Fig. 3c), where the area-weighted mean lies slightly outside the PDF maximum, showing how a great number of small glaciers are presenting higher losses. On the other hand, a clearer relationship between the glacier slope (computed here as the lowermost 20% altitudinal range slope) and glacier-wide MB arises, with steeper glaciers having less negative glacier-wide MBs (Fig. S6 and S9). Glaciers with a gentle tongue slope generally present longer response times and higher

ice thickness, which are associated with more negative mass balances (Hoelzle et al., 2003; Huss and Fischer, 2016; Zekollari et al., 2020). These results are in agreement with the findings by Fischer et al. (2015), who computed the geodetic mass balance of all the Swiss glaciers for the 1980-2010 period. Overall, the topographical relationships found here are similar, although more negative than for the Swiss Alps (Huss, 2012; Huss et al., 2015), showing how the southernmost glaciers in the Écrins and

5 Vanoise regions present stronger glacier mass losses. This is mostly due to their mediterranean climatic influence compared to the more continental Swiss and Austrian glaciers, which results in more negative MB in a warming climate (Oerlemans and Reichert, 2000). Nonetheless, results from this type of bivariate analysis can show rather biased trends, since the topographical variables are highly intercorrelated, with for example small glaciers having steeper slopes and *vice versa* (Gardent et al., 2014). The position and evolution of the equilibrium line can totally reverse the trends of small or steep glaciers, so these relationships

can strongly vary depending on the region or time period observed.

### 3.4 Comparison with previous studies and observations

In order to put into perspective the reconstructions presented in this study, we compare them to an updated version from the Marzeion et al. (2015) reconstructions (B. Marzeion, personal communication, October 2019 - January 2020), and to all the available glacier-wide MB observations and remote sensing estimates in the French Alps. The goal of this comparison is not

to draw conclusions on the quality of either reconstruction, but to analyse the differences among them and to try to understand the causes. In the updated version of Marzeion et al. (2015) - referred as $M_{15U}$ from now on - a global MB model relying on temperature and solid precipitation was used to reconstruct MB time series for all the glaciers in the world present in the Randolph Glacier Inventory (Consortium, 2017). This model was optimized based on five parameters: the temperature sensitivity of the glacier (local); and a precipitation correction factor, precipitation lapse rate, temperature threshold for solid

precipitation and melt temperature threshold (global). As in Bolibar et al. (2020b), the approach by $M_{15U}$ was cross-validated respecting the spatiotemporal independence in order to evaluate its performance for unobserved glaciers and years. Due to the highly different methodologies and forcings of the two models, a direct comparison is not possible, so the following analysis is focused on the overall trends and sensitivities in the reconstructions and their potential sources. All the specific differences and details between the two models can be found in Sect. 2 from the Supplement.

The annual variability (Fig. 6), driven by climate, is quite similar between the two reconstructions. Conversely, important differences are found for different subperiods in the amplitude of the area-weighted mean glacier-wide MB series. These differences are the greatest in the 1970s, 1980s and 2010s, with similar average values for the 1990s and 2000s (Fig. 6 and S7). $M_{15U}$ presents less negative and more positive glacier-wide MB values in the 1970s, but on the contrary, it presents more negative values in the 1980s compared to our results. We believe there might be two potential reasons for this: (1) In 1976

there was a shift in the winter mass balance regime in the French Alps, with more humid winters bringing more accumulation; and in 1982 there was a shift in the summer mass balance, resulting in increased ablation (Thibert et al., 2013). Since both models use parameterized or statistical relationships for MB response to precipitation and temperature, they are likely to react differently to these changes. A similar situation is found from the year 2003 onwards, where there was a substantial increase in temperatures and mass loss (e.g. Six and Vincent, 2014). Our reconstructions show a marked change in 2003 (change of slope

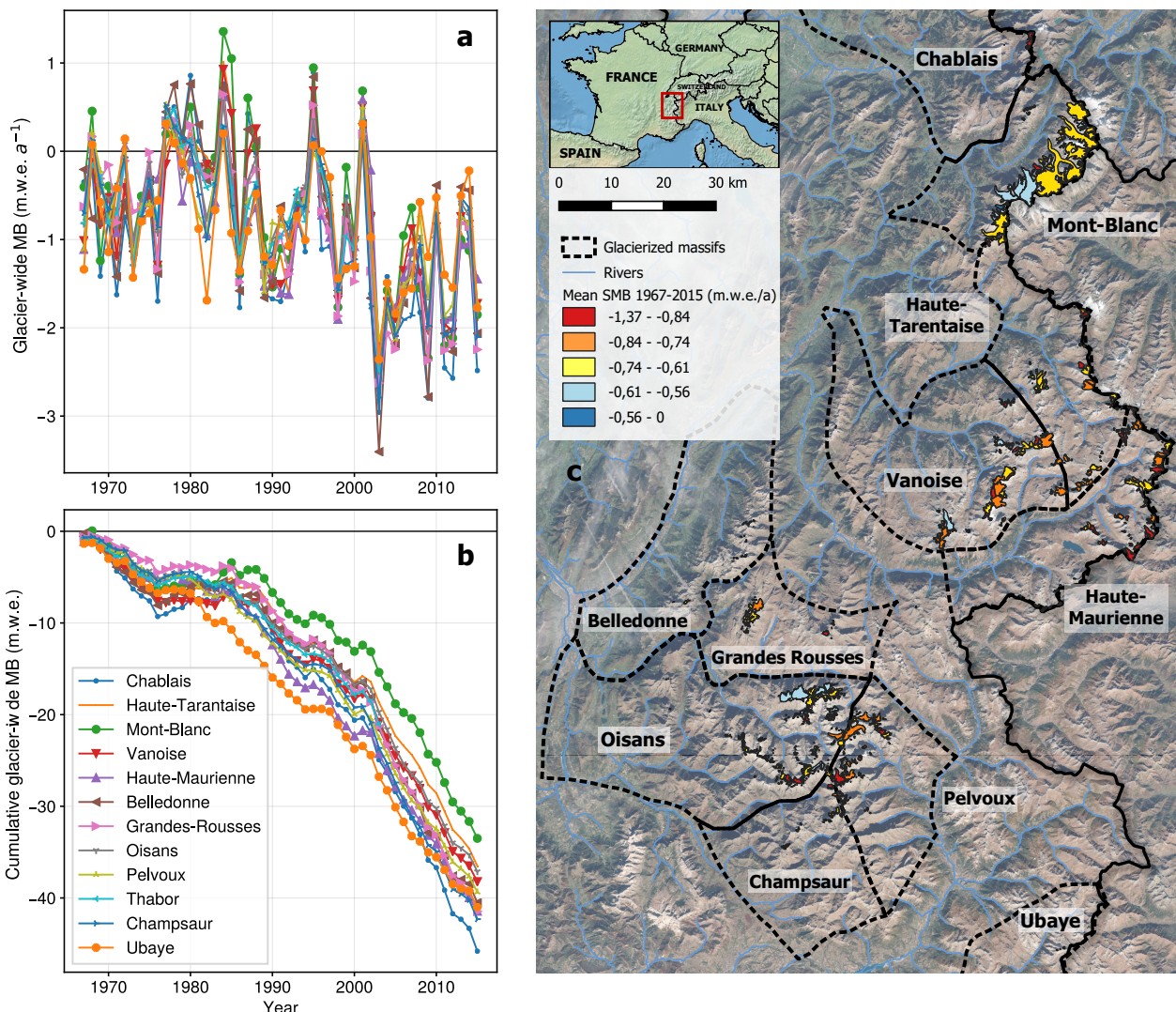

**Figure 5.** (a) Averaged annual glacier-wide MB and (b) cumulative averaged glacier-wide MB time series for each of the massifs in the French Alps between 1967 and 2015. (c) Glacierized massifs in the French Alps with the average glacier-wide MB for the 1967-2015 period. Coordinates of bottom left map corner: 44°32' N, 5°40' E. Coordinates of the top right map corner: 46°08' N, 7°17' E. (Basemap © imagico.de)

in the cumulative plot in Fig. 6), whereas $M_{15U}$ present a rather linear trend. The fact that $M_{15U}$ used a volume-area scaling compared to the interpolated topographical data from inventories from this study means that the topographical feedback of the models might differ as well throughout the reconstructed period. (2) For the 1967-1983 interval, the amount of available glacier-wide MB data for training is much lower than for the rest of the period (green numbers in Fig. 6). This is likely the

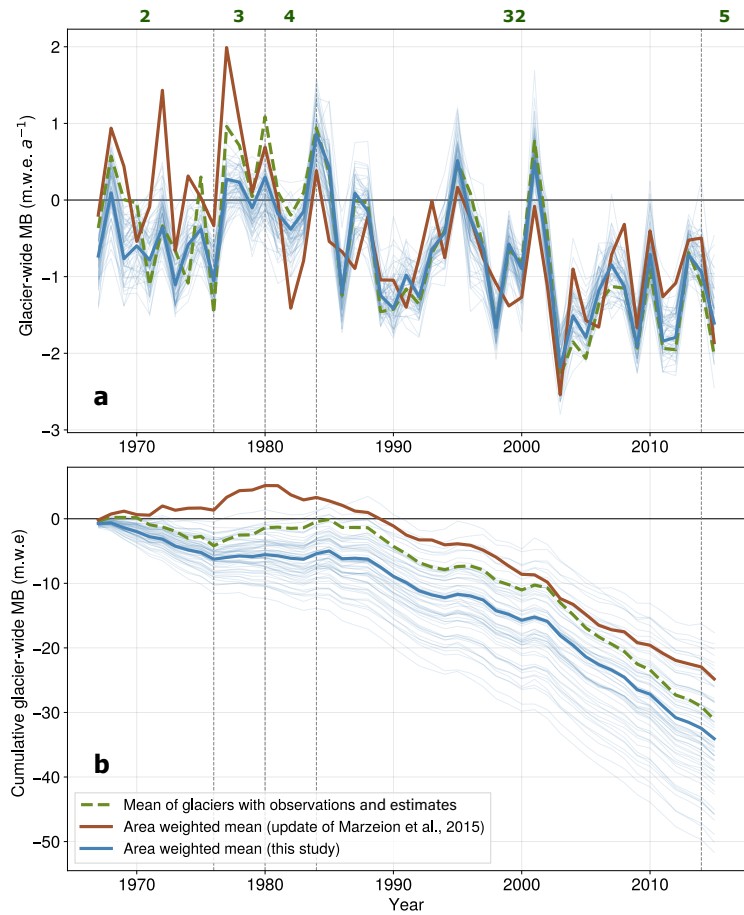

**Figure 6.** Comparison of (a) annual and (b) cumulative glacier-wide MB simulations in the French Alps between this study, reconstructions from an update from Marzeion et al. (2015) and the mean of all observations and remote sensing estimates available in the French Alps. Green numbers indicate the number of glaciers with MB observations and remote sensing estimates for each period and thin light blue lines indicate the area-weighted mean of each of the cross-validation ensemble members.

reason why the differences between our reconstructions and training data are greater for that period (Fig. 6). On the other hand, the similarities between our reconstructions and the training data for the 1984-2014 period are explained by the fact that the 32 glaciers with observations represent around 45% of the total glacierized area in the French Alps in the year 2003. For the periods before and after this interval, differences and uncertainties in the reconstructed values are greater because of the smaller

5  sample size.

In the following, we argue that similarities between observations, remote sensing estimates and the reconstructed glacier-wide MB values for the 1984-2015 period in this study (Fig. 6) are not due to overfitting. First, for the vast majority of the 661 French glaciers, the reconstructions are based on an ensemble of cross-validated models, which intrinsically limits overfitting

(see Sect. 2). Second, we analysed the deviation to the climatological mass-balance signal of the MB for each cluster of glacier-sizes. This analysis is presented in Sect. 3 of the supplementary material. It reveals that the similarities between the training data and the reconstructed glacier-wide MB values for the 1984-2015 period in Fig. 6 originate from big glaciers, that dominate both in the area-weighted reconstructions and in the training data (Fig. S3 and S4). However, for the other glacier-size classes, our reconstruction shows different patterns from the data in the training data, which suggests that the model is not overfitting (Fig. S3).

## 4 Data availability

The full glacier-wide MB dataset is available in the following Zenodo repository: https://doi.org/10.5281/zenodo.3925378 (Bolibar et al., 2020a).

## 5 Conclusions

We presented a dataset of annual glacier-wide MB of all the glaciers in the French Alps (44° - 46°13'N, 5.08° - 7.67°E) for the 1967-2015 period (Bolibar et al., 2020a). This dataset has been reconstructed using deep learning (i.e. an artificial neural network), based on direct and remote sensing annual glacier-wide MB observations and estimates, climate reanalysis and topographical data from multitemporal glacier inventories. The deep learning model is capable of reconstructing glacier-wide MB time series for unobserved glaciers in the same region based on patterns and structures learnt by the artificial neural network from the training data and their relationships with predictors. An extensive cross-validation was implemented to understand the characteristics of the MB signal in the region and to assess the method's validity and uncertainty. The average accuracy (RMSE) of the dataset is estimated at 0.55 m.w.e. $a^{-1}$ with an explained variance ($r^2$) of 75%. Reconstructions show a mean area-weighted glacier-wide MB of -0.69±0.21 (1 $\sigma$) m.w.e. $a^{-1}$ for the 1967-2015 period. Important differences are found among different massifs, with the Mont-Blanc (-0.68 m.w.e. $a^{-1}$), Oisans (-0.75 m.w.e. $a^{-1}$ both) presenting the lowest mass losses and the Chablais (-0.93 m.w.e. $a^{-1}$), Champsaur (-0.86 m.w.e. $a^{-1}$) and Haute-Maurienne and Ubaye (-0.84 m.w.e. $a^{-1}$ both) showing the highest losses. In order to put these results into perspective, this reconstruction was compared to all available glacier-wide MB observations and remote sensing estimates in the French Alps as well as the physical/empirical reconstructions from another study (update from Marzeion et al., 2015). Interesting differences were found between the two methods, highlighting the different sensitivities and responses of different approaches to climate shifts that occurred during the study period. These differences are particularly relevant in the 1970s and 1980s, previous to a winter precipitation and summer temperature shift that occurred in the French Alps in the years 1976 and 1982, respectively. Moreover, after the famous 2003 European heatwave, glaciers experienced an acceleration in mass loss which is well captured by our reconstruction. This open glacier-wide MB dataset can be useful for hydrological or ecological studies in need of net glacier mass contributions of glacierized catchments in the French Alps. The publication of such open datasets is essential to future community-based data-driven scientific studies.

## 6 Code availability

The source code of ALPGM v1.2 is accessible at https://github.com/JordiBolibar/ALPGM, with its DOI: 10.5281/zenodo.3922935 (Bolibar, 2020).

*Author contributions.* All authors contributed to writing and editing the manuscript. JB performed the simulations, processed the data and plots and performed the analysis. AR provided the surface mass balance remote sensing data and contributed to the glaciological analysis, IG participated in the climate and regional analysis and CG contributed to the statistical aspects of the methods.

*Competing interests.* The authors declare that they do not have any competing interests.

*Acknowledgements.* The authors would especially like to thank Ben Marzeion for providing the updated dataset of glacier-wide SMB reconstructions and for the interesting discussions and feedback. Thomas Condom gave insightful comments on the manuscript and Christian Vincent provided feedback to improve the discussion in the glaciological analysis. This work would not be possible without the work of all the people involved in the more than 60 years of glaciological measurements in the French Alps by the French National Observation Service GLACIOCLIM (https://glacioclim.osug.fr/). We are very grateful for the review comments from Matthias Huss and Ben Marzeion that helped to improve the overall clarity and quality of the paper.

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
