# Peer review of "A deep learning reconstruction of mass balance series for all glaciers in the French Alps: 1967-2015"

_Earth System Science Data, 2020_

## Referee Comment (RC1) · Anonymous Referee #1 · 25 Mar 2020

This paper present a reconstruction of the surface mass balance (SMB) of all French glaciers for the period 1967-2015 based on an deep learning (DL) approach. It strongly relates to the study on the methodology recently published by the same authors in The Cryosphere. The data set is comprehensive, interesting and certainly deserves publication. However, there are presently some weaknesses in the presentation of the findings, as well as in the validation of the approach.

Substantial comments:

- Structure: The paper needs to be restructured. A Data section is needed – this is presently merged into the Methods. I would also suggest that the study site characteristics are presented in a separate section. At present this is contained in the Results section. The Methods section needs a clearer structure, separating between the actual approach and the uncertainty assessment.

- A major problem of the paper as it stands now is, in my opinion, the lack of validation with independent measurements: The training data set for 32 glaciers is based on "remote-sensing" (Rabatel et al., 2016). As this lies the foundation to the entire study, more effort should be invested to describe this data set (methods, uncertainties). The training data set also contains models and assumptions – the annual SMB of these glaciers has not been directly measured. This needs to be emphasized. Measured (!) information on SMB is available from two sources: (1) the direct glaciological surveys, (2) geodetic surveys.

Although (1) is probably included in the training data set, it should explicitly be shown (e.g. figure with cumulative SMBs) how well the DL approach reproduces the observed SMB series. This would give direct evidence on the performance of the approach, independent from the training data set by Rabatel et al. (2016) that also includes model assumptions.

I was surprised to see that the study does not show any direct validation with data on geodetic mass balance. With the annual resolution of the data set presented here, this would be straight-forward to achieve. For many glaciers in the French Alps, geodetic mass balances over varying time periods are available (see e.g. data base of the WGMS). I assume that some of them have already been used in the set up of the training data set, but probably not all of them. Geodetic mass balances would yield a fantastic way to validate the DL-based estimates of regional variability in mass balances. New remote sensing data sets (e.g. TanDEM-X in combination with SRTM, or ASTER) would also allow computing geodetic mass balances for each individual glacier. I see that deriving such a new geodetic data set is beyond the scope of this study, but it is indispensable to compile the available geodetic survey for French glaciers in this regional assessment in order to gain confidence in the results.

- Mass balance modelling: Deep learning and mass balance modelling are compli-
mentary approaches. Whereas, the introduction gives relatively little details about the
potential of mass balance modelling (and previous studies for the French glaciers) to
reconstruct SMB (e.g. Vincent 2002), much detail is given towards the end of the text
when comparing the present results to those by Marzeion et al. (2015). Also it was not
clear to me why the authors decided to only rely on the results of this relatively coarse
study (global scale, no specific calibration to the French Alps) and not more recent and
more detailed model results on the past SMB of glaciers in the French Alps by Zekol-
lari et al. (2019) (regional scale, high-resolution climate forcing data, glacier-specific
calibration based on geodetic mass balances).

- Language: Although the paper is well-written in general, there are several instances
where the writing could be improved (e.g. "allows to . . .", p2, line 18 and other in-
stances, is not correct English). Proof-reading by a native English speaker would cer-
tainly help.

Detailed comments:

- Page 1, line 8: cross-validation against which data set? Please clarify here.

- Page 1, line 16: I would not refer to "meltwater contributions" here: Annual values of
glacier-wide SMB do not actually yield "meltwater" but just annual glacier mass loss.
Meltwater has a strong seasonal component and also occurs in the case of SMB=0 or
SMB >0! - Page 2, line 12: order references according to year

- Page 2, line 13: Digital Elevation Model, instead of "maps", is typically used

- Page 2, line 22: This is also true for various other global glacier models (see Hock et al
2019 for a compilation). All of these models provide annual SMB for French glaciers in
the PAST (although with probably limited skill). See also comment above. The detailed
regional modelling study including the French Alps (Zekollari et al 2019) should also be
mentioned.

[Figure]

- Page 2, line 28: It seems strange to refer to the results of the present paper already within the paper itself. It is clear that the data set is already available, but it should not be referred to, as this is where it is actually described.

- Figure 1: Although it is stated in the caption that the figure is schematic, it leaves a wrong impression on the density of available information for training the DL approach: In fact, green lines should only make up for 5% of the glaciers, whereas the figure implies that it is more than a third. It should be revised accordingly.

- Page 3, line 10: Although this paper is closely related with TC paper of the same authors, the location of the training data sets needs to be presented here.

- Page 3, line 16: It is excellent that topographical information is available in repeated time steps throughout the study period. However, it remains unclear how this updated topographical data was included in the DL approach. This is quite relevant information as the feedback of retreating glaciers (shrinking area, changes in area-elevation distribution and terminus position) exerts an important effect on glacier-wide SMB. A detailed description of this is required.

- Page 4, line 1: Similar comment as above: I would just refer to the paper where the model is described and not have separate references to the model and the publication. You can state in the Data availability section where the code of the model is located.

- Page 4, line 15: It is a quite relevant aspect in my opinion (probably worth mentioning in the Intro) that the DL approach (e.g. in comparison to in-situ observations and mass balance modelling) only provides annual SMB, but no information on seasonal mass balance as well as mass balance gradients. If my understanding is wrong, please correct me. But these additional variables are crucial for various aspects of impact assessment and model development.

- Figure 2/5: Please add letters (A/B) to the panels

- Page 7, line 18: For a more recent reference on exactly this topic, please also see

Zekollari et al. (2020), Geophysical Research Letters

- Page 8, line 2: Please remove the reference to Huss&Hock, 2015. It does not fit here in my opinion.

- Page 8, line 31: see also substantive comment above: To me, this appears too long and too detailed. The comparison to a modelling study is a nice addition but it does not actually allow evaluation of the present results. More focus needs to be put on the validation using fully independent field observations (in-situ) and geodetic surveys.

- Page 10, line 3: Potentially also related to the way glacier retreat (updating of area-elevation distribution) is accounted for the in model by Marzeion et al 2015, and the present approach? Probably worth to discuss here as well.

- Page 11, line 6: Actually, all Supplementary Figures should be referenced from the main text. I found the analysis in the Supplementary interesting but not straightforward to understand. It might be beneficial to present this additional analysis more prominently in the main text.

- Page 12, line 12: Is there no possibility to go beyond the year 2015 by the way? In my understanding the trained DL approach should enable to predict mass balance also for the most recent years. This would be quite interesting as the last years were extraordinary in terms of their mass losses.

---

## Referee Comment (RC2) · Ben Marzeion (Referee) · 4 May 2020

Bolibar et al. present the results of a new approach to reconstruct glacier mass balances at times and/or locations where meteorological conditions (and some topographical information) are known, but no observations of glacier mass balance exist. Their approach, based on a neural network algorithm, adds considerable diversity to the existing group of reconstruction methods. The thorough validation of the results leads to great confidence in the robustness of the method.

Except for some minor issue listed below, the manuscript is very clear and easy to follow. The data set produced and presented here will be of great use for the community.

[Figure]

I particularly appreciate the great care that has been taken in documenting the test for overfitting in the supplementary material.

I recommend publication once the authors have gone through the list of questions/suggestions below.

Specific/minor comments:

- P1 L9: please specify "1 \sigma" instead of "\sigma" for clarity

- P1 L10: the "moderately" should only apply to the 1980s, I think

- P1 L10: avoid line break within negative number

- P1 L12: unclear, what "this period" refers to

- abstract: why are no uncertainties given for the values of the different massifs? (also concerns the conclusions)

- P2 L8: "these points" refers to the points of MB measurements, but this reference is not very clear here; also, it's not the points that show nonlinear variability, but the measurements at the points; suggest to rephrase

- P2 L23: there more four global parameters in the Marzeion et al. (2012) model, and I wouldn't necessarily say they were "optimized", because that "optimization" was very subjective...

- Fig. 1: the figure certainly works well for presentations etc., but I'm not sure it is necessary here, since the text describes very well what is done, and there is little to be gained from the figure.

- P3 L14-15: it would be great if you can add a sentence or two here, specifying how any difference in the altitude of the glaciers' centroids and the reanalysis grid points were treated (lapse rates or similar?)

- P4 L22 or lower: It might be worth pointing out/discussing that the density of observations used in the LOGO cross validation is denser towards the end of the reconstruction interval, when presumably, also the quality of the meteorological data are higher, such that the uncertainty of the methods might be underestimated for the (roughly) first half of the period. I also wonder if/how this interferes with your assessment of the model's ability to reconstruct the more neutral MB values during 1967-1984?

- Fig. 2: since there are so many lines, it is somewhat hard to see the distribution. Particularly in the lower panel, a histogram for showing the distribution of the accumulated values (vertically, to the right of the panel) would be quite interesting. It would be possible to see, e.g., how/if the area weighted mean differs from the "ensemble" mean and/or median, if the distribution is (a)symmetric, etc. Just a suggestion to consider.

- Fig. 3: why are no uncertainties included for the decadal averages?

- Fig. 4: great figure! But a bit busy (just visually); would it be possible to mute the background image a bit (and then perhaps change the text color to black) so that the colors of the glaciers stand out more?

- P8 L16: it's more than three parameters: one local (the temperature sensitivity) and four global ones (precipitation correction factor, precipitation lapse rate, temperature threshold for solid precipitation, and melt temperature threshold); see Figs. 4-7 in Marzeion et al. (2012)

- P8 L22: perhaps clarify that the 38 glaciers are not the global sample used for calibration

- P8 L31: I believe that the CV results in the Marzeion et al. (2012) study are also influenced by the global "optimization" (see above) of the four parameters; probably, a focus on the Alps would have led to a different parameter choice, and hence different CV results.

- P10 L1 and following: another reason for the different behavior around the 2003 "break point" might be that the Marzeion et al. (2012) model, by construction, cannot

capture the lasting effect that the extreme 2003 year may have had on albedo; while your model may be able to capture this (I guess – I'm not sure) by essentially taking the time as an additional predictor?

- Fig. S2: would it be possible to re-arrange the legend such that it is easier to compare the "B" to the "M" lines (e.g., shift the lowest line in the legend to the right)?

---

## Author Comment (AC1) · 13 May 2020

Authors reply to Anonymous reviewer 1 on "A deep learning reconstruction of mass balance series for all glaciers in the French Alps: 1967–2015"

**Anonymous reviewer 1**

**1 Substantial comments**

This paper presents a reconstruction of the surface mass balance (SMB) of all French glaciers for the period 1967-2015 based on a deep learning (DL) approach. It strongly relates to the study on the methodology recently published by the same authors in The Cryosphere. The data set is comprehensive, interesting and certainly deserves publication. However, there are presently some weaknesses in the presentation of the findings, as well as in the validation of the approach.

We are grateful for the overall positive comments by the reviewer. We believe the comments highlighted some aspects that ought to be clearer, and served to further develop some analysis to increase the quality of the paper. All comments have been answered, including the changes made in the manuscript, presented in bold to distinguish them from the unchanged sentences in the updated sections.

Some relevant improvements have been done to the methods during the review process. We have trained a new cross-validation ensemble of 60 members and updated the dataset results. This new ensemble is based on weighted bagging (Hastie et al., 2009) of Leave-Some-Years-and-Glaciers-Out cross-validation (Bolibar et al., 2020), which balances the training data in the model in order to better take into account the lack of data between 1967-1983. The main results and conclusions have not changed, only leading to a slightly less negative average mass balance (from -0.72 to -0.71 m w.e. $a^{-1}$), and slightly higher uncertainties due to the increased presence of underrepresented values of the 1967-1983 period (RMSE: from 0.49 to 0.55 m w.e. $a^{-1}$ and $r^2$: from 0.79 to 0.75). We believe this even more rigorous cross-validation leads to more accurate results and uncertainty estimations.

**1.1 Structure**

The paper needs to be restructured. A Data section is needed – this is presently merged into the Methods. I would also suggest that the study site characteristics are presented in a separate section. At present this is contained in the Results section. The Methods section needs a clearer structure, separating between the actual approach and the uncertainty assessment.

The methods section has been restructured following the reviewer's suggestions. Sect. 2 is now called "Data and methods", and it includes three sections: 2.1 Data: with a brief introduction of the French Alps and the datasets used and their coverage; 2.2 Methods: with an explanation of the methods used to reconstruct the annual glacier-wide SMB values; and 2.3 Uncertainty assessment: with the analysis of the method's uncertainties and bias.
* * *
**1.2 SMB validation**

A major problem of the paper as it stands now is, in my opinion, the lack of validation with independent measurements: The training data set for 32 glaciers is based on "remote-sensing" (Rabatel et al., 2016). As this lies the foundation to the entire study, more effort should be invested to describe this data set (methods, uncertainties). The training data set also contains models and assumptions – the annual SMB of these glaciers has not been directly measured. This needs to be emphasized. Measured (!) information on SMB is available from two sources: (1) the direct glaciological surveys, (2) geodetic surveys.

Although (1) is probably included in the training data set, it should explicitly be shown (e.g. figure with cumulative SMBs) how well the DL approach reproduces the observed SMB series. This would give direct evidence on the performance of the approach, independent from the training data set by Rabatel et al. (2016) that also includes model assumptions.

I was surprised to see that the study does not show any direct validation with data on geodetic mass balance. With the annual resolution of the data set presented here, this would be straight-forward to achieve. For many glaciers in the French Alps, geodetic mass balances over varying time periods are available (see e.g. data base of the WGMS). I assume that some of them have already been used in the set up of the training data set, but probably not all of them. Geodetic mass balances would yield a fantastic way to validate the DL-based estimates of regional variability in mass balances. New remote sensing data sets (e.g. TanDEM-X in combination with SRTM, or ASTER) would also allow computing geodetic mass balances for each individual glacier. I see that deriving such a new geodetic data set is beyond the scope of this study, but it is indispensable to compile the available geodetic survey for French glaciers in this regional assessment in order to gain confidence in the results.
* * *
There are two main ways to validate the results of a model: comparing them to another independent dataset (e.g., the geodetic MB dataset that the reviewer refers to), and applying cross-validation. The use of cross-validations ensures a true out-of-sample validation, allowing the validation of the full dataset. This presents a substantial advantage when few data are available, as all data can be used both for training/calibration and validation. For the case of spatio-temporal data, this needs to be carefully done, as it was discussed in detail in Bolibar et al. (2020). The reconstructed annual glacier-wide SMB series were not validated against other datasets than the ones mentioned in the paper since the vast majority of available data in the region has already been used for training. The dataset from Rabatel et al.

(2016) is extremely useful in (1) the fact that its uncertainties are very close from uncertainties from the glaciological method (0.35±0.06 m w.e. a$^{-1}$), and (2) the fact that it is calibrated from geodetic mass balances, meaning that the geodetic data explicitly serve to calibrate and validate the bias.

Regarding the validation against glaciological observations, this has been done as part of the cross-validation in the uncertainty assessment. As we have mentioned in different instances of the "Data and methods" section, all details regarding the methods can be found in Bolibar et al. (2020), which is a purely methodological paper. Since the methods in Bolibar et al. (2020) were based on a case study using the very same 32 glaciers used in this study, it means that the methods and cross-validation results are exactly the same as the ones presented in detail in that paper. It also includes a lot of information regarding the dataset of these 32 glaciers, the performance for each glacier and year, as well as detailed plots with the comparison of simulations and observations (Fig. 6 to 10) . Therefore, our intention with the present paper submitted to ESSD is to apply the methods from Bolibar et al. (2020) in order to generate a regional dataset. Since all the details regarding the methods can be found in a separate paper, here we prefer to focus on the results and the conclusions rather than repeating what has already been presented in detail elsewhere.

On the other hand, we agree that there are a few, independant geodetic mass balances for shorter periods available, which can be used to validate the bias for a sub-period of the reconstructions, but its added value is lower than that of a cross-validation over the entire reconstruction period. However, following the reviewer's suggestions, we have compared the Pléiades geodetic mass balance data from Berthier et al. (2014) and the ASTER geodetic mass balance data from the newly published Davaze et al. (2020) to our reconstructions. Since these two studies cover different sub-periods, the comparisons have been done separately. Both studies cover only the beginning of the 21$^{st}$ century, so the relevance of these bias validation is moderate, as our model has been calibrated to reconstruct SMB for over 50 years, with different climate conditions, especially before and after the summer heatwave of the year 2003.

A new section analyzing this ("1 Comparison with independent geodetic mass balance data"), including the two following figures have been added to the supplementary material, in order to illustrate this.

"1 Comparison with independent geodetic mass balance data

All available annual glacier-wide SMB data in the French Alps have been used to train the SMB ANN of the present study. However, some multi-annual geodetic mass balance (MB) datasets exist that can provide a means to validate the reconstruction's bias for specific glaciers during multi-annual time intervals. This type of analysis is more limited than the cross-validation done to annual glacier-wide SMB values in Bolibar et al. (2020), as it only gives information about the bias of a sub-period of the reconstructions instead of the accuracy found via cross-validation. Our SMB reconstructions are compared against ASTER

Authors reply to Anonymous reviewer 1 on "A deep learning reconstruction of mass balance series for all glaciers in the French Alps: 1967–2015"

geodetic MB from Davaze et al. (2020) for the 2000-2015 and 2003-2012 periods (Fig. S1 and S2) and against Pléiades geodetic MB from Berthier et al. (2014) for the 2003-2012 period (Fig. S2).

For certain glaciers, the ASTER and Pléiades geodetic MB give slightly less negative MB than the glaciological SMB used to train the deep learning SMB model. This fact might explain the slightly more negative trend of our reconstructions seen for the 2000-2015 and 2003-2012 periods, which experienced very negative SMB after the well known summer 2003 heatwave. This is quite surprising, since both the GLACIOCLIM glaciological SMB measurements and the annual glacier-wide SMB data from Rabatel et al. (2016) have been calibrated with geodetic MB from optically-derived DEMs, which have a very high spatial resolution. Overall, the independent geodetic MB are well within the uncertainty range of our model. There are some exceptions for specific glaciers in the Mont-Blanc massif, such as Bossons, Talèfre and Tour. These glaciers have very large and high altitude accumulation areas, not seen in almost any glacier in our training dataset. On the other hand, for most of the mid-sized glaciers the reconstructions show a good agreement."

[Figure]

Authors reply to Anonymous reviewer 1 on "A deep learning reconstruction of mass balance series for all glaciers in the French Alps: 1967–2015"

[Figure]

**1.3 Language**

Although the paper is well-written in general, there are several instances where the writing could be improved (e.g. "allows to : : :", p2, line 18 and other instances, is not correct English). Proof-reading by a native English speaker would certainly help.

The text has been revised again, and any remaining grammar issues will hopefully be fixed by Copernicus' language correction services at publication.

**2 Detailed comments**

Page 1, line 8: cross-validation against which data set? Please clarify here.

The sentence in the abstract has been updated as suggested by the reviewer.

"**The method's validity was assessed through an extensive cross-validation against a dataset of 32 glaciers**, with an estimated average error…."

Page 1, line 16: I would not refer to "meltwater contributions" here: Annual values of glacier-wide SMB do not actually yield "meltwater" but just annual glacier mass loss. Meltwater has a strong seasonal component and also occurs in the case of SMB=0 or SMB >0!

Indeed, the annual glacier-wide SMB is the net annual mass change of the glacier, so it is not precise to refer to it as meltwater contributions. The sentence has been updated accordingly.

"…provides relevant and timely data for studies in the fields of glaciology, hydrology and ecology in the French Alps, in need of regional or glacier-specific **annual net glacier mass changes** in glacierized catchments."
* * *
- Page 2, line 12: order references according to year.
* * *
The references have been updated as suggested.
* * *
Page 2, line 13: Digital Elevation Model, instead of "maps", is typically used
* * *
The acronym has been updated as suggested.
* * *
Page 2, line 22: This is also true for various other global glacier models (see Hock et al 2019 for a compilation). All of these models provide annual SMB for French glaciers in the PAST (although with probably limited skill). See also comment above. The detailed regional modelling study including the French Alps (Zekollari et al 2019) should also be mentioned.
* * *
Indeed, any global past SMB simulations include the European Alps with the French Alps, but these two specific studies were chosen since they were dedicated publications on the European Alps (Huss 2012, Marzeion et al., 2012) and they covered the full period from this study (1967-2015).

The study from Zekollari et al. (2019) was not used for comparison as the main purpose of their paper was to present the future evolution of the glaciers. The study covers the past period between 2003 until 2017, but this is a minimal fraction of time period of our study. It was not clear to us if annual glacier-wide SMB data is available for the 1967-2015 period, as the period seems to only be covered during validation, so it might only include glaciers with WGMS observations. Nonetheless, since all studies available use substantially different climatic forcings and SMB modelling approaches to our study, the type of comparison would still be the same, only serving to showcase the different sensitivities and responses of models to the past climate in the French Alps.

We have updated the sentence in order to give some context as suggested by the reviewer:

"On the other hand, SMB reconstructions have already been carried out in the European Alps, **providing a basis for comparison between different approaches (see Hock et al.**

**(2019) for a compilation). Two studies include reconstructions in the European and thus the French Alps over a substantial period of the recent past**: ..."
* * *
Page 2, line 28: It seems strange to refer to the results of the present paper already within the paper itself. It is clear that the data set is already available, but it should not be referred to, as this is where it is actually described.
* * *
We understand that this might seem strange, and this would probably not be done in another journal. But the instructions for manuscript preparation of ESSD say:

"**Data sets**: The data sets described in the manuscript need to be deposited in reliable data repositories including the assignment of digital object identifiers. Authors are required to properly cite the data sets in the abstract, text, and the reference list (see section References below)"

Therefore, since here we are referring to the dataset itself, which is different from the paper (which acts only as a presentation), we believe that the citation makes sense according to the author guidelines. Nonetheless, if the reviewer and editor think this is not the correct way to use dataset citations for this journal we will remove it.
* * *
Figure 1: Although it is stated in the caption that the figure is schematic, it leaves a wrong impression on the density of available information for training the DL approach: In fact, green lines should only make up for 5% of the glaciers, whereas the figure implies that it is more than a third. It should be revised accordingly.
* * *
Indeed, the representation was not accurate. The number of glaciers with observations has now been reduced in Fig. 1. Nonetheless, it does not exactly account for 5% (as it would leave only one glacier which would not help to convey the message), but the representation is much more accurate now.
* * *
Page 3, line 10: Although this paper is closely related with TC paper of the same authors, the location of the training data sets needs to be presented here.
* * *
The methods section of this paper is a brief summary of the whole Bolibar et al. (2020) paper. We believe that the exact location of the 32 training glaciers is not very relevant for the reader to understand the methods used. The most important fact regarding the location is the fact that the training glaciers are distributed along most of the glacierized massifs of the French Alps, thus presenting a representative spatial coverage. As previously mentioned in other comments, for all the details regarding the methods the reader can refer to the dedicated methodological paper. Our intention is to avoid repeating unnecessary information, and to allow the reader to quickly understand the methods and the implications on the results without bothering too much with the details.

Authors reply to Anonymous reviewer 1 on "A deep learning reconstruction of mass balance series for all glaciers in the French Alps: 1967–2015"

Following the reviewer's suggestion, and in order to make the paper fully self-sufficient, we have included the map of the study area with the glaciers used for training in the supplementary material.

"For the reconstruction presented here, a dataset of 32 French alpine glaciers has been used for training, covering most of the massifs within the French Alps, which exhibit a great variability of topographical characteristics **(Fig. S10)**."

[Figure]

Figure S10. French Alpine glaciers used for model training and validation and their classification into three clusters or regions (Écrins,Vanoise, Mont-Blanc). Coordinates of bottom left map corner: 44º32' N, 5º40' E. Coordinates of the top right map corner: 46º08' N, 7º17'E.

Page 3, line 16: It is excellent that topographical information is available in repeated time steps throughout the study period. However, it remains unclear how this updated topographical data was included in the DL approach. This is quite relevant information as the feedback of retreating glaciers (shrinking area, changes in area-elevation distribution and terminus position) exerts an important effect on glacier-wide SMB. A detailed description of this is required.

We extended the comments in this section in order to make the presentation of the training predictors of the model clearer.

"Out of the 32 glaciers from this dataset, four glaciers include direct SMB measurements from the GLACIOCLIM observatory, some of which between 1949 and 2018 (Vincent et al., 2017) and 28 glaciers include estimates of annual glacier-wide SMB from remote sensing between 1984 and 2014 (Rabatel et al., 2016). **This dataset, with a total of 1048 annual glacier-wide SMB values, is used as a reference. Unlike point SMB, glacier-wide SMB is influenced by both climate and topography, producing complex interactions between climate and glacier morphology which need to be taken into account in the model. For each annual glacier-wide SMB value available, the following data are compiled to train the ANN with an annual time step**: (1) climate data from the SAFRAN meteorological reanalyses (Durand et al., 2009) with: cumulative positive degree days (CPDD), cumulative winter snowfall, cumulative summer snowfall, mean monthly temperature and mean monthly snowfall, all variables being quantified at the altitude of the glacier's centroid; and **(2) annually interpolated topographical data between the 1967, 1985, 2003 and 2015 glacier inventories in the French Alps (Gardent et al., 2014), with: mean and maximum glacier altitude, slope of the lowermost 20% altitudinal range of the glacier, surface area, latitude, longitude and aspect. Therefore, the topographical feedback of the shrinking glaciers is captured from these annually interpolated topographical predictors.** These parameters were identified as relevant for glacier-wide SMB modelling in the French Alps (Bolibar et al., 2020) and the dates of the glacier inventories determined the time interval for the reconstructions presented here."

Page 4, line 1: Similar comment as above: I would just refer to the paper where the model is described and not have separate references to the model and the publication. You can state in the Data availability section where the code of the model is located.

Bolibar et al. (2020) is not a presentation of the model, but a presentation of the deep learning SMB modelling approach, with a case study on the French Alps. Therefore, when referring to the model itself (software), we prefer to use the citation for the model, as it is more precise. The Data availability section already includes both the dataset and the model's source code.
* * *
**Page 4, line 15: It is a quite relevant aspect in my opinion (probably worth mentioning in the Intro) that the DL approach (e.g. in comparison to in-situ observations and mass balance modelling) only provides annual SMB, but no information on seasonal mass balance as well as mass balance gradients. If my understanding is wrong, please correct me. But these additional variables are crucial for various aspects of impact assessment and model development.**
* * *
We agree that this aspect is important. This is mentioned in different sections of the manuscript, in the abstract, introduction and in some sections. We have tried to clarify this by systematically referring to the reconstructed SMB as "annual glacier-wide SMB". Moreover, in the update from the reviewer's comment on Page 3 Line 16, we have contrasted this fact with the characteristics of point mass balance data. If the reviewer thinks it would be clearer to refer to this in another way than "annual glacier-wide SMB" we could adapt the manuscript accordingly.

Seasonal mass balances are indeed very useful for several applications (see for instance Viani et al., 2018 in the field of hydrology). However in our case, the use of annual glacier-wide SMB data is not a problem, since ALPGM uses the delta-h (Huss et al., 2010) parameterization in order to redistribute the annual glacier-wide mass changes along the glacier. This geometry update is only used for glacier evolution simulations, which have nothing to do with the dataset presented here, so this information has been omitted in the methods from the present study.
* * *
**Figure 2/5: Please add letters (A/B) to the panels.**
* * *
Letters have been added to the different panels of Figures 2 and 5.
* * *
**Page 7, line 18: For a more recent reference on exactly this topic, please also see Zekollari et al. (2020), Geophysical Research Letters**
* * *
Indeed. A reference has been added to this paper.
* * *
**Page 8, line 2: Please remove the reference to Huss&Hock, 2015. It does not fit here in my opinion.**
* * *
Well spotted. This is in fact a LaTex reference typo. The intended reference was: Huss et al. (2015): "New long-term mass-balance series for the Swiss Alps", Journal of Glaciology. The citation has been updated accordingly.
* * *
Page 8, line 31: see also substantive comment above: To me, this appears too long and too detailed. The comparison to a modelling study is a nice addition but it does not actually allow evaluation of the present results. More focus needs to be put on the validation using fully independent field observations (in-situ) and geodetic surveys.
* * *
In order to make this section lighter and more straight to the point, we have moved a whole paragraph which compared and detailed the differences between both models to the Supplementary material. This allows to convey the message that a direct comparison is not possible, jumping straight to results and conclusions and leaving the technical details for the avid reader who will be willing to read the Supplementary material.

The new section S2 is the following one:

"2 Model differences between the updated version of Marzeion et al. (2015) and this study

In order to contrast the results from Sect. 3.4, three important different aspects between our approach and the one of M15U need to be highlighted:

1. $M_{15U}$'s model works with simplified physics, with a temperature-index model calibrated on observations; in this study we used a fully statistical approach based on deep learning, where physics-based considerations only appear in the predictor selection.

2. $M_{15U}$ calibrated their model with SMB observations of 38 glaciers, most of them located in Switzerland for the 1901-2013 period; in this study we used observations of 32 glaciers, all located in the French Alps for the 1967-2015 period.

3. $M_{15U}$ forced their updated model with CRU 6.0 (update of Harris et al., 2014), with 0.5° latitude/longitude grid cells, which has a significantly lower spatial resolution and suitability to mountain areas than the SAFRAN reanalysis (Durand et al., 2009) used in this study, in which altitude bands and aspects are considered for each massif, and meteorological observations from high-altitude stations are assimilated.

The cross-validations of both studies determined a performance with an average RMSE of 0.66 m.w.e. a$^{-1}$ and an r$^2$ of 0.43 for $M_{15U}$ for the European Alps, and an average RMSE of 0.49 m.w.e. a$^{-1}$ and an r$^2$ of 0.79 for this study. However, due to the highly different methodologies and forcings of the two models, a direct comparison is not possible, so the following analysis is focused on the overall trends and sensitivities in the reconstructions and their potential sources."
* * *
Page 10, line 3: Potentially also related to the way glacier retreat (updating of area elevation distribution) is accounted for in the model by Marzeion et al 2015, and the present approach? Probably worth discussing here as well.

That's a very good point. On top of the differences in the climate forcings, there might be differences in the topographical feedback of the models due to the different modelling approaches. This element has been added to the discussion:

Authors reply to Anonymous reviewer 1 on "A deep learning reconstruction of mass balance series for all glaciers in the French Alps: 1967–2015"

"The fact that $M_{15U}$ used a volume-area scaling compared to the interpolated topographical data from inventories from this study means that the topographical feedback of the models might differ as well throughout the reconstructed period."
* * *
Page 11, line 6: Actually, all Supplementary Figures should be referenced from the main text. I found the analysis in the Supplementary interesting but not straightforward to understand. It might be beneficial to present this additional analysis more prominently in the main text.
* * *
All Supplementary Figures that were not previously mentioned in the main text have now been added as references in the appropriate sections. With this we hope the reader will be encouraged to read the supplementary material in case she/he is interested in the detailed methods. By giving the references in the right context, it is now easier to relate the explanations of Sect. 2 in the Supplementary to the content of the main text. Our intention is to keep the methods section and technical details as light as possible, in order to convey an easy message based on the results, and allow the avid reader to check the details in the methods paper and the Supplementary material.
* * *
Page 12, line 12: Is there no possibility to go beyond the year 2015 by the way? In my understanding the trained DL approach should enable to predict mass balance also for the most recent years. This would be quite interesting as the last years were extraordinary in terms of their mass losses.
* * *
Indeed, it would be very interesting, but unfortunately with the current framework it cannot be done, unless some modifications are done to it. Since we are working with topographical data by interpolating glacier inventories, we would need a glacier inventory after 2015 in order to have topographical information for these years. One possible hypothesis to bypass this would be to continue interpolating the topographical data with the same trend as for the 2003-2015 period, but this would introduce some inhomogeneity in the method. On the other hand, the version of the meteorological reanalysis that we are using (SAFRAN) only extends until 2016. A new version has just been released, that covers the same period until 2019 ; its use would require some technical adaptations and reprocessing of the years previous 2015 for homogeneity. Considering these two facts, we believe it is not worth the time investment for just four additional years.

---

## Author Comment (AC2) · 13 May 2020

Authors reply to Ben Marzeion on "A deep learning reconstruction of mass balance series for all glaciers in the French Alps: 1967–2015"

**Ben Marzeion**

**1 General comments**

Bolibar et al. present the results of a new approach to reconstruct glacier mass balances at times and/or locations where meteorological conditions (and some topographical information) are known, but no observations of glacier mass balance exist. Their approach, based on a neural network algorithm, adds considerable diversity to the existing group of reconstruction methods. The thorough validation of the results leads to great confidence in the robustness of the method. Except for some minor issue listed below, the manuscript is very clear and easy to follow. The data set produced and presented here will be of great use for the community. I particularly appreciate the great care that has been taken in documenting the test for overfitting in the supplementary material. I recommend publication once the authors have gone through the list of questions/suggestions below.

We are grateful for the positive and encouraging comments. These comments will help improve the manuscript's quality and clarity. Most figures in the paper have been re-processed taking into account the feedback, hopefully leading to better visualization and presentation. Every comment/suggestion has been addressed individually in the following section.

As explained in one of the comments regarding the validation approach based on cross-validation, we have trained a new cross-validation ensemble of 60 members and updated the dataset results. This new ensemble is based on weighted bagging (Hastie et al., 2009) of Leave-Some-Years-and-Glaciers-Out cross-validation (Bolibar et al., 2020), which balances the training data in the model in order to better take into account the lack of data between 1967-1983. The main results and conclusions have not changed, only leading to a slightly less negative average mass balance (from -0.72 to -0.71 m w.e. $a^{-1}$), and slightly higher uncertainties due to the increased presence of underrepresented values of the 1967-1983 period (RMSE: from 0.49 to 0.55 m w.e. $a^{-1}$ and $r^2$: from 0.79 to 0.75). We believe this even more rigorous cross-validation leads to more accurate results and uncertainty estimations.

**2 Specific/minor comments**

P1 L9: please specify "1\sigma" instead of "\sigma" for clarity.

The sentence in the abstract has been updated as suggested by the reviewer.

P1 L10: the "moderately" should only apply to the 1980s, I think

Authors reply to Ben Marzeion on "A deep learning reconstruction of mass balance series for all glaciers in the French Alps: 1967–2015"

Indeed. The sentence has been updated:

"We estimate an average regional area-weighted glacier-wide SMB of -0.72±0.20 (1\sigma) m.w.e. $a^{-1}$ for the 1967-2015 period, **with negative mass balances** in the 1970s (-0.52 m.w.e. $a^{-1}$), **moderately negative** in the 1980s (-0.12 m.w.e. $a^{-1}$), and an increasing negative trend from the 1990s onwards, up to -1.39 m.w.e. $a^{-1}$ in the 2010s."
* * *
**P1 L10: avoid line break within negative number**
* * *
This has been fixed with the rephrasing of some parts of the abstract.
* * *
**P1 L12: unclear, what "this period" refers to**
* * *
The sentence has been updated to clearly indicate the time period:

"Following a topographical and regional analysis, we estimate that the massifs with the highest mass losses **for the 1967-2015** period are the..."
* * *
**abstract: why are no uncertainties given for the values of the different massifs? (also concerns the conclusions)**
* * *
Because we have no way to dissociate the uncertainties for each massif from the overall uncertainties computed through cross-validation. Therefore, all massifs would display the same uncertainty, which is already given with the average performance of the method for this region (RMSE = 0.55 m w.e. $a^{-1}$). If the reviewer thinks it would still be better to give the uncertainty for each massif we can add it in the abstract and text.
* * *
**P2 L8: "these points" refers to the points of MB measurements, but this reference is not very clear here; also, it's not the points that show nonlinear variability, but the measurements at the points; suggest to rephrase**
* * *
This sentence has been adapted to improve clarity as suggested:

"**These different point SMB measurements** can show a high nonlinear variability..."

Authors reply to Ben Marzeion on "A deep learning reconstruction of mass balance series for all glaciers in the French Alps: 1967–2015"
* * *
P2 L23: there more four global parameters in the Marzeion et al. (2012) model, and I wouldn't necessarily say they were "optimized", because that "optimization" was very subjective...
* * *
The word "optimized" has been removed to erase these connotations from the sentence as suggested by the reviewer:

"They used a minimal model relying only on temperature and precipitation data, based on a temperature-index method, with two parameters to calibrate the temperature sensitivity and the precipitation lapse rate."
* * *
Fig. 1: the figure certainly works well for presentations etc., but I'm not sure it is necessary here, since the text describes very well what is done, and there is little to be gained from the figure.
* * *
Indeed, the main key aspects of the overall analysis are already given in the abstract and in the text. Nonetheless, we believe it is a complementary way to show the regional variability, as it shows in a single figure the spread of glacier behaviour and the common variability in a nice and easy way. This is our personal opinion, if the reviewer strongly suggests to remove it, we will move it to the supplementary material.
* * *
P3 L14-15: it would be great if you can add a sentence or two here, specifying how any difference in the altitude of the glaciers' centroids and the reanalysis grid points were treated (lapse rates or similar?)
* * *
The explanation on climate data and predictors has been updated with the following sentences in order to give some context on how the forcings are adjusted to each glacier centroid's altitude:

"(1) climate data from the SAFRAN meteorological reanalyses (Durand et al., 2009) with: cumulative positive degree days (CPDD), cumulative winter snowfall, cumulative summer snowfall, mean monthly temperature and mean monthly snowfall, all variables being quantified at the altitude of the glacier's centroid. **In order to capture the climate signal at each glacier's centroid, temperatures are taken from the nearest SAFRAN 300 m altitudinal band and adjusted with a 6ºC/km lapse rate. The updated temperature is then used to update the snowfall amount from the same 300 m altitudinal band.**"
* * *
P4 L22 or lower: It might be worth pointing out/discussing that the density of observations used in the LOGO cross validation is denser towards the end of the reconstruction interval, when presumably, also the quality of the meteorological data are higher, such that the uncertainty of the methods might be underestimated for the (roughly) first half of the period. I also wonder if/how this interferes with your assessment of the model's ability to reconstruct the more neutral MB values during 1967-1984?
* * *
That's a very good point. That was one of our main concerns during the validation process, which we tried to address in two different ways.

First of all, we performed a separate cross-validation with only data from the 1967-1984 period, in order to specifically assess the performance during this period. This is explained in the newly created Sect. "2.3 Uncertainty assessment" (as suggested by Anonymous reviewer 1). This was already present in the version of the manuscript sent for review.

On the other hand, in order to improve our estimates and to better take into account this lack of homogeneity in the dataset, we have trained a new ensemble of models based on Leave-Some-Years-and-Glaciers-Out (LSYGO) cross-validation, as explained in Bolibar et al. (2020). We used an ensemble of 60 CV models using weighted bagging (Hastie et al., 2009) by giving +33% more weight to data between 1967-1984, in order to compensate for this lack of observations during this period, which covers a third of the 49-year period. This has not affected much the results, and the conclusions remain exactly the same, but it allows giving a more accurate and realistic assessment of the model's performance, with a RMSE of 0.55 m w.e. $a^{-1}$, a coefficient of determination of 0.75 and an average bias of -0.019 m w.e. $a^{-1}$.
* * *
Fig. 2: since there are so many lines, it is somewhat hard to see the distribution. Particularly in the lower panel, a histogram for showing the distribution of the accumulated values (vertically, to the right of the panel) would be quite interesting. It would be possible to see, e.g., how/if the area weighted mean differs from the "ensemble" mean and/or median, if the distribution is (a)symmetric, etc. Just a suggestion to consider.
* * *
That is a good idea. A panel to the right of the cumulative plot has been added with a histogram, the PDF and the position of the area weighted mean SMB.

Authors reply to Ben Marzeion on "A deep learning reconstruction of mass balance series for all glaciers in the French Alps: 1967–2015"

[Figure]

Fig. 3: why are no uncertainties included for the decadal averages?

Fig. 3 has been updated with decadal uncertainties.

Authors reply to Ben Marzeion on "A deep learning reconstruction of mass balance series for all glaciers in the French Alps: 1967–2015"

[Figure]

Fig. 4: great figure! But a bit busy (just visually); would it be possible to mute the background image a bit (and then perhaps change the text color to black) so that the colors of the glaciers stand out more?

We do agree that Fig. 4 could be a little bit overwhelming. We have updated it following the suggestions of the reviewer. Now the colours of the glaciers are more visible, and it has a more homogenous feeling with all contour lines in black.

Authors reply to Ben Marzeion on "A deep learning reconstruction of mass balance series for all glaciers in the French Alps: 1967–2015"

[Figure]

P8 L16: it's more than three parameters: one local (the temperature sensitivity) and four global ones (precipitation correction factor, precipitation lapse rate, temperature threshold for solid precipitation, and melt temperature threshold); see Figs. 4-7 in Marzeion et al. (2012).

The sentence has been updated with the correct information as it follows:

"This model was optimized based on five parameters: the temperature sensitivity of the glacier (local); and a precipitation correction factor, precipitation lapse rate, temperature threshold for solid precipitation and melt temperature threshold (global)"

P8 L22: perhaps clarify that the 38 glaciers are not the global sample used for calibration.

The new section in the supplementary material has been updated following this suggestion:

Authors reply to Ben Marzeion on "A deep learning reconstruction of mass balance series for all glaciers in the French Alps: 1967–2015"

"**M15U calibrated their model with global SMB observations, including 38 glaciers in the European Alps, most of them located in Switzerland for the 1901-2013 period**; in this study we used observations of 32 glaciers, all located in the French Alps for the 1967-2015 period."

P8 L31: I believe that the CV results in the Marzeion et al. (2012) study are also influenced by the global "optimization" (see above) of the four parameters; probably, a focus on the Alps would have led to a different parameter choice, and hence different CV results.

Indeed, that is what we tried to convey with the warning to the readers. We are comparing a global model with a regional model, so the specificity of the calibration is completely different, giving a clear advantage to the regional model. We hope that with the updated sentence from the previous comment this will be more clear to the reader.

P10 L1 and following: another reason for the different behaviour around the 2003 "break point" might be that the Marzeion et al. (2012) model, by construction, cannot capture the lasting effect that the extreme 2003 year may have had on albedo; while your model may be able to capture this (I guess – I'm not sure) by essentially taking the time as an additional predictor?

Our mass balance model does not have any perception of time, as no time stamps are used as predictors. I believe the main reason(s), as stated in the article, are the fact that we use higher resolution climate forcings, which better capture the climate signal on the glaciers, and most importantly, that the deep learning SMB model is nonlinear, which gives it a greater deal of flexibility to simulate this kind of transitions compared to linear models. This was already observed during the cross-validation analysis in Bolibar et al. (2020), where the linear model with Lasso, which behaves similarly to a temperature-index model, showed biases at the beginning and end of the 1984-2015 period, as the parameters were calibrated to fit the whole period, which presents rather neutral SMBs at the beginning and strongly negative SMBs by the end (see the Figure below taken from Bolibar et al., 2020). The nonlinear deep learning SMB model showed much lower biases, further demonstrating that the climate and glacier systems are highly nonlinear.

Authors reply to Ben Marzeion on "A deep learning reconstruction of mass balance series for all glaciers in the French Alps: 1967–2015"

[Figure]

Fig. S2: would it be possible to re-arrange the legend such that it is easier to compare the "B" to the "M" lines (e.g., shift the lowest line in the legend to the right)?

The legend in Fig. S4 (previously S2) has been updated in order to have the "B" and "M" lines on different columns.

---

## Author Response (AR2)

Authors reply Matthias Huss on "A deep learning reconstruction of mass balance series for all glaciers in the French Alps: 1967–2015"

**Matthias Huss**

The authors have carefully revised their manuscript which has substantially improved and is almost ready for acceptance in my opinion. The additional work performed strengthens the conclusions of the article. However, some issues remain that should be addressed by the authors. Some of them have already been mentioned in my first review. I thus just copy in my former comment that I would like the authors to answer.

We are very grateful for the feedback provided. The remarks have pointed out interesting elements of the paper that needed improvements. We have addressed all the specific comments raised by the reviewer in detailed answers.

Following the comparison with independent geodetic MB suggested by the reviewer, we have taken this opportunity to recalibrate 2000-2015 MB series for a few glaciers outside the training dataset. Glaciers with low uncertainties (< 0.15 m.w.e. $a^{-1}$) in the ASTER-derived MB that can as well be considered as outliers for our MB model can benefit from this recalibration (particularly large and steep glaciers in the Mont-Blanc massif). We have adapted a paragraph explaining this in Sect. "2.3 Uncertainty assessment", as well as in Sect S1 from the Supplementary. The recalibrated estimates have also been added to the new Fig. 2 (as suggested), in order to illustrate the effects of this recalibration for certain glaciers. The overall average results do not change, only with an ever so slightly less negative MB for the 1967-2015 from -0.71 to -0.69 m.w.e. $a^{-1}$.

"In order to further validate the reconstructions presented here, a comparison against independent ASTER (Davaze et al., 2020) and Pléiades (Berthier et al., 2014) geodetic MB data was performed, that helps to assess the bias of the MB reconstructions for the 2000-2015 (Fig. 2) and 2003-2012 (Fig. S2) sub-periods. The photogrammetric geodetic MB used to calibrate the MB datasets from Rabatel et al. (2016) and the glaciological observations from GLACIOCLIM have a much higher resolution than ASTER-derived geodetic MB, but the comparison can bring interesting information for glaciers outside the training dataset. Our reconstructions show a good agreement with the geodetic MB for certain regions (e.g. Grandes Rousses), except for some particular steep large high-altitude glaciers (e.g. Bossons and Taconnaz in the Mont-Blanc massif) that substantially differ from most glaciers in the French Alps. A more detailed analysis and additional figures comparing the MB datasets can be found in the supplementary material. In order to exploit this additional geodetic MB dataset, we have recalibrated our MB reconstructions for the 2000-2015 period using the ASTER-derived geodetic MB from Davaze et al. (2020) for some glaciers outside our training dataset (i.e. B20' in Fig. 2). Since ASTER-derived geodetic MB present important uncertainties for small glaciers (i.e. < 1 km2), we have only recalibrated MB series for 16 large glaciers outside the training dataset with uncertainties lower than 0.15 m.w.e. $a^{-1}$. The calibration has been performed by adding the average annual bias between Davaze et al. (2020) and this study for the 2000-2015 sub-period."

Another small improvement added is the presentation of the MB reconstructions dataset in two different formats: (a) a new single netCDF file, for easy-to-access data

management and processing and (b) the previously presented multiple file CSV dataset. The Zenodo repository has been updated with these new files.

"A major problem of the paper as it stands now is, in my opinion, the lack of validation with independent measurements: The training data set for 32 glaciers is based on "remote-sensing" (Rabatel et al., 2016). As this lies the foundation to the entire study, more effort should be invested to describe this data set (methods, uncertainties). The training data set also contains models and assumptions – the annual SMB of these glaciers has not been directly measured. This needs to be emphasized. Measured (!) information on SMB is available from two sources: (1) the direct glaciological surveys, (2) geodetic surveys.

You are right in pointing this out. We apologize for not having revised this in more detail, and we acknowledge this fault.

All references to MB observations have been updated, in order to refer to "direct MB observations" (for the four GLACIOCLIM glaciers), and "MB remote sensing estimates", for the 28 glaciers from Rabatel et al. (2016).

The methods have been extended in our previous revision, with a dedicated section explaining the training data, briefly introducing neural networks and explaining the cross-validation approach.

Although (1) is probably included in the training data set, it should explicitly be shown (e.g. figure with cumulative SMBs) how well the DL approach reproduces the observed SMB series. This would give direct evidence on the performance of the approach, independent from the training data set by Rabatel et al. (2016) that also includes model assumptions.

In the new "Data" section we have explicitly mentioned that the neural network is trained with direct glaciological observations.

We have added a new figure comparing the MB series of Mer de Glace, Argentière, Saint Sorlin and Gebroulaz (the four GLACIOCLIM glaciers) with the results of our MB reconstructions. Two versions of the MB reconstructions are compared: (1) A cross-validation version (ANN CV) in order to show the out-of-sample performance, and (2) the version used in the reconstructions (ANN) where the neural network has actually seen the data of the GLACIOCLIM glaciers. This comparison shows how the deep learning MB ensemble is capable of extrapolating to other unseen glaciers, and for the glaciers for which it has seen the data, it captures the interannual variability with a great degree of accuracy.

Authors reply Matthias Huss on "A deep learning reconstruction of mass balance series for all glaciers in the French Alps: 1967–2015"

[Figure]

The description of the basis of the principal training data set (Rabatel et al., 2016) is still very limited (see page 3, line 16). The reader just knows that it is based on "remote sensing". However, it is clear that remote sensing alone is (yet) unable to provide annual glacier-wide mass balance. It should be described which approaches and assumptions went into the training data set and that glacier-wide mass balance is not measured, as in the case of the four GLACIOCLIM series, but is based on modelling optimally constrained with geodetic mass balances and snowline observations. I do not at all want to imply that the data set is unsuitable or inaccurate but it should be honestly stated that these are not actually "observations" as the annual mass balances are referred to several times. Related to this, I would also ask the authors to follow my advice of comparing the annual (!) series from their approach against the four GLACIOCLIM data sets. This is directly measured data and would allow investigating whether DL is able to capture these observed year-to-year variations.

We agree that the description of the methods from Rabatel et al. (2016) is important for the reader to understand the type of data used to train the deep learning model. We have added an explanation of how the annual remote sensing estimates are obtained in the Data section.

"Out of the 32 glaciers from this dataset, four glaciers include direct MB measurements from the GLACIOCLIM observatory, some of which since 1949, which have been calibrated using photogrammetric geodetic MB (Vincent et al., 2017). On the other hand, 28 glaciers include estimates of annual glacier-wide MB from remote sensing between 1984 and 2014 (Rabatel et al., 2016). These remote sensing estimates were computed using (1) the end-of-summer snowline for every year, which in the European Alps is a proxy of the equilibrium-line altitude (ELA); and (2) geodetic MB for the 1984-2014 period quantified from two high-resolution DEMs. Both data sources are used to reconstruct the annual glacier-wide MB of each individual glacier for the same period of the geodetic MB."

Regarding the comparison of the MB reconstructions to glaciological observations, we definitely agree that this information is relevant to validate the methods. As previously mentioned, a new figure has been added with this information. We initially intended to strongly base the methods on our previous paper published this year in *The Cryosphere*, but we agree that constantly referring the reader to another paper makes the task of following the paper much more complicated.
* * *
What is shown on top of page 4 is not actually Data. It rather seems to be a description of the method to incorporate them. Please try to better separate data used in the approach from the methods. I realise though that this is not trivial to achieve here.
* * *
Indeed, the separation here is not clear at all. We could have opted to just mention the datasets used for training and move the explanations on what variables are actually used for training in the "Methods" section. We agree that this is possibly the best way to proceed for most cases, but for a machine learning model, data means a lot more than just what datasets are used. Since we believe it is much easier for the reader to read what datasets are used and then what variables are taken for training, we have decided to rename the "Data" section to "Training data". We believe this title is more representative of what the section actually is: a description of both the datasets and the actual data and predictors used for the model. By keeping this section together, the reader can easily link the mentioned datasets to the explanations on what actual variables are used for training, which is a key element for machine learning models.
* * *
A general comment (sorry, for not bringing this up already in the first review): Throughout the paper mass balance is referred to as SMB (surface mass balance). But does this approach actually compute SURFACE mass balance? And not rather the total mass balance of a glacier (see Cogley et al., 2011, for an overview)? The approach is trained on the Rabatel-dataset which depends on geodetic balances for constraining the mass loss. Geodetic mass balance covers both surface mass balance and basal/internal balance. Consequently, I would consistently omit the "surface" when you refer to mass balance. Anyway, this is just terminology, as it is

clear that the difference between SMB and total mass balance will be minimal for Alpine glaciers.

This is a very good comment, and luckily a very easy one to fix, so we are glad it was brought up. Indeed, all data used for training here are mass balance and not surface mass balance. Since both the glaciological observations and the equilibrium-line altitude remote sensing estimates are constrained with geodetic mass balances, this means that all data are actually mass balances. We have updated all references in the manuscript text and figures (except for the explanation of the glaciological measurements) in order to accurately refer to this with the appropriate term.

It is a pity that the new validation with independent geodetic mass balances has not made it into the main paper. I think it would strongly support the credibility of the data set and is important enough to not only be presented in the Supplementary.

We have decided to follow this advice and to add the comparison with independent geodetic MBs in the "2.3 Uncertainty assessment" section as Fig. 2. We have only added the 2000-2015 comparison though, since it has more glaciers and a longer time period.

[revised manuscript text omitted]

---

## Author Response (AR3)

**Reinhard Drews**

We are grateful for the additional feedback that has helped to improve some last aspects of the manuscript. All minor corrections have been added to the text, and the two comments have been addressed and answered.
* * *
Comment 1 comparison to interannual Glacioclim data: My understanding is that model training was done using annual MB only (e.g., p4, l. 20). If so, does this mean that the models has never seen the interannual Glacioclim time series? Please state this explicitly, as this matters for the evaluation of Fig. S1. Also, state in the caption of S1 that this is an inter-annual time series. This Figure is quite isolated and poorly linked to the main text (with one sentence on p5 l 15). This was one of the major concerns during review, and although I see that it has been taken into account, the takeway message is still muddled.

We have updated all instances of the word "inter-annual" to "annual" in order to avoid confusion with the word "seasonal". With "annual" we mean one single value per year. The reference to Figure S1 comparing the MB reconstructions to the GLACIOCLIM glaciological observations has been extended, in order to give a better context to the reader and extract some more conclusions from it.
* * *
Comment 2 structure/presentation: Don't use sentence like "..can be found in the supplementary information." Let the reader know exactly which section you refer to, so that he/she can quickly find it. During another re-read also have an eye un unnecessarily convoluted sentences (e.g. p3, l14) which are better expressed in multiple shorter sentences.

All instances of this have been updated following the editor's suggestions. Some long sentences have been also shortened in order to improve readability.

[revised manuscript text omitted]